# Next-Generation Antisense Oligonucleotide of TGF-β2 Enhances T Cell-Mediated Anticancer Efficacy of Anti-PD-1 Therapy in a Humanized Mouse Model of Immune-Excluded Melanoma

**DOI:** 10.3390/cancers14215220

**Published:** 2022-10-25

**Authors:** Hong Kyu Lee, Cho-Won Kim, Dohee Ahn, Ryeo-Eun Go, Youngdong Choi, Kyung-Chul Choi

**Affiliations:** 1Laboratory of Biochemistry and Immunology, College of Veterinary Medicine, Chungbuk National University, Cheongju 28644, Korea; 2Division of Endocrinology, Children’s Hospital Boston, Harvard Medical School, Boston, MA 02115, USA

**Keywords:** melanoma, TGF-β2, anti-PD-1, antitumor immunity, humanized mouse

## Abstract

**Simple Summary:**

Upregulation of transforming growth factor-beta (TGF-β) signaling in melanoma contributes to the formation of immune-suppressive tumoral environments and is associated with poor response to immunotherapeutics, including anti-programmed death-1 (PD-1) therapy. Our study aimed to investigate the immunotherapeutic potential of a novel next-generation TGF-β2 antisense oligonucleotide (ngTASO) in combination with anti-PD-1 therapy using a melanoma-bearing human immune system mouse model. Our findings confirm that blockade of TGF-β signaling by ngTASO improves the T cell-mediated antitumor potential of anti-PD-1 therapy via facilitating intratumoral infiltration of CD8+ cytotoxic lymphocytes and their activation. These results suggest that ngTASO is a promising combination strategy with anti-PD-1 therapy for the treatment of immune-excluded melanoma.

**Abstract:**

Anti-programmed death-1 (PD-1) immunotherapy is one of the most promising therapeutic interventions for treating various tumors, including lung cancer, bladder cancer, and melanoma. However, only a subset of patients responds to anti-PD-1 therapy due to complicated immune regulation in tumors and the evolution of resistance. In the current study, we investigate the potential of a novel transforming growth factor-beta2 (TGF-β2) antisense oligonucleotide (ngTASO), as a combination therapy with an anti-PD-1 antibody in melanoma. This study was conducted in a melanoma-bearing human immune system mouse model that recapitulates immune-excluded phenotypes. We observed that the TGF-β2 blockade by ngTASO in combination with PD-1 inhibition downregulated the tumor intrinsic β-catenin, facilitated the infiltration of CD8+ cytotoxic lymphocytes (CTLs) in the tumor, and finally, enhanced the antitumor immune potentials and tumor growth delays. Blockade of TGF-β2 combined with PD-1 inhibition also resulted in downregulating the ratio of regulatory T cells to CTLs in the peripheral blood and tumor, resulting in increased granzyme B expression. In addition, co-treatment of ngTASO and anti-PD-1 augmented the PD-L1 expression in tumors, which is associated with an improved response to anti-PD-1 immunotherapy. These results indicate that the combination of ngTASO and anti-PD-1 exerts an enhanced T cell-mediated antitumor immune potential. Hence, co-inhibition of TGF-β2 and PD-1 is a potentially promising immunotherapeutic strategy for immune-excluded melanoma.

## 1. Introduction

Programmed death-1 (PD-1, also known as CD279) is an immune checkpoint protein expressed on the surface of T cells and is responsible for the suppression of the antitumor immune response by binding to the programmed death ligand-1 (PD-L1, also known as CD274 or B7 homolog 1) on tumor cells [1]. PD-1/PD-L1 signaling promotes tumor-induced immunosuppression through impeding T cell receptor-mediated production of interleukin (IL)-2 and proliferation of T cells [2,3]. Clinically, most advanced cancers, including melanoma, non-small cell lung cancer, and gastric cancer, utilize PD-1/PD-L1 signaling to achieve tumor escape from immune surveillance [4,5,6]. Therefore, targeting the PD-1/PD-L1 axis is currently one of the most promising strategies with clinical significance in the immune-oncology field. The major anticancer drugs approved by the Food and Drug Administration or in clinical trials for immune checkpoint inhibition are associated with PD-1/PD-L1. Among these PD-1/PD-L1 blockers, pembrolizumab (an anti-PD-1 antibody) is known to be one of the most potent immune checkpoint inhibitors (ICIs) and is approved for the treatment of metastatic melanoma in certain situations [7]. However, the therapeutic benefits of a PD-1/PD-L1 blockade by pembrolizumab in melanoma are only achieved in a specific subset of patients [8]. These varying outcomes may result from the complexity of the tumor microenvironments (TME), including T cell depletion, T cell dysfunction, and altered PD-L1 expression in tumors [9]. Based on the mechanisms involved in resistance to anti-PD-1 monotherapy, studies on various targets for improving the immunotherapeutic potential of PD-1 inhibition are in progress. 

The blockade of the transforming growth factor-beta (TGF-β) pathway may be one of the most attractive strategies in malignant melanoma, providing a prospect of combination therapy with PD-1 inhibition through enhancing T cell activation and infiltration in tumors [10,11]. TGF-β plays a crucial role in the pathogenesis of melanoma. It is produced from components of the TME, including tumor cells, stroma cells, and immune cells [12,13]. Although TGF-β has a tumor suppressive effect in the premalignant state of cancer, an increase in TGF-β with tumor progression is commonly associated with poor prognosis in melanoma [13,14]. The possible mechanisms of TGF-β-mediated tumor progression are numerous and may involve the following: (1) promotion of neo-angiogenesis [15], (2) increased invasion and motility through upregulation of epithelial—mesenchymal transition signaling [16,17], (3) inflammation in some tumors [18], and (4) escape from the immune surveillance [19,20]. In particular, TGF-β suppresses the antitumor immunity through induction of suppressive immune cells such as regulatory T cells (Tregs) and myeloid-derived suppressor cells, as well as suppression of immune cells such as cytotoxic T lymphocytes (CTLs) and natural killer (NK) cells [12,20,21]. Recently, we confirmed the antitumor immune potential of TGF-β2 inhibition by an antisense oligonucleotide (ASO) and suggested the possibility of using it in combination with an immunostimulator (IL-2) in the breast cancer and melanoma models [22,23]. However, the effects of the TGF-β signaling blockade by ASO therapy in combination with PD-1 inhibition on antitumor immunity against melanoma have not been investigated yet.

In the current study, we investigate the enhanced tumor growth inhibition and related antitumor immune mechanisms of anti-PD-1 antibody in combination with a novel next-generation ASO of TGF-β2 (ngTASO) for the treatment of melanoma. In the melanoma-bearing human immune system (HIS) mouse model, we observed that a combination of ngTASO and anti-PD-1 antibody additionally delay the tumor growth, which is mediated by the potentiation of T cell-mediated antitumor immunity such as activation and increased infiltration of CTLs. These results indicate that blocking TGF-β2 by ngTASO in combination with PD-1 inhibition is considerably implicated in enhancing antitumor immunity, and provide a scientific rationale for its development as a promising anticancer strategy for melanoma.

## 2. Materials and Methods

### 2.1. Oligonucleotide

ngTASO was developed by Autotelic Bio, Inc. (Seongnam, Korea). ngTASO was designed to be human TGF-β2-specific, and the sequences of ngTASO are 5′-GGCGGCATGTCTATTTTGTA-3′ modified with 2′-methoxyethyl as indicated by underlined letters, and full phosphorothioate (PS) backbone modification. 

### 2.2. Cell Cultures

The A2058 cell line (human melanoma) was purchased from the American Type Culture Collection and cultured in a humidified chamber with 5% CO_2_ at 37 °C in DMEM supplemented with 10% FBS, 100 IU/mL penicillin, and 100 μg/mL streptomycin. For qRT-PCR and Western blot, A2058 cells were seeded at an approximate density of 2.4 × 10^5^ cells/well in a six-well plate using the complete medium. After 24 h of incubation, cells were treated with ngTASO without the transfection reagent for 24 h (for qRT-PCR) or 48 h (for Western blot).

### 2.3. qRT-PCR and Western Blot

For qRT-PCR analysis, total RNA from the cell culture was extracted using the Qiagen Rneasy kit (Qiagen, Venlo, The Netherlands) following the manufacturer’s instructions. qRT-PCR was achieved using the QuantStudio 3 Real-time PCR device (Applied Biosystems, Waltham, MA, USA). The primer sequences of target genes used in the current study are presented in Table 1. The expression level of the *TGFB2* gene was normalized to serine and arginine rich splicing factor 9 (*SRSF9*) by the ΔΔ cycle threshold (ΔΔ Ct) method as described previously [24]. Quantification of target gene expression was measured by the formula:

ΔCt = Ct (*TGFB2* gene) − Ct (*SRSF9* gene)

The relative expression level (ΔΔCt) was obtained by comparing ΔCt treated sample to the ΔCt control. Relative fold change to the control group was calculated by the formula (2^−ΔΔCt^).

### 2.4. Experimental Animals

Female NOD/scid/IL-2Rγ^−/−^/B2m^−/−^ (NOD.Cg-*B2m^tm1Unc^ Prkdc^scid^ Il2rg^tm1Wjl^*/SzJ, NSG-B2m) mice were purchased from The Jackson Laboratory (Bar Harbor, ME, USA). Animal studies were performed using 7-week-old mice at the Laboratory Animal Research Center of Chungbuk National University under specific pathogen-free conditions. All the procedures of animal experiments were approved by the Institutional Animal Care and Use Committee (IACUC) of the Chungbuk National University (CBNUA-1350-20-01).

### 2.5. Generation of a Melanoma-Bearing Human Immune System Mouse Model

The human melanoma-bearing humanized mice were generated by transplanting human PBMCs (hu-PBL NSG-B2m model) followed by subcutaneous inoculation of A2058 cells, applying a previously described protocol with slight modifications [23]. Briefly, after stabilization of human PBMCs obtained from Zen-Bio (Zen-Bio, INC., Research Triangle, NC, USA), 1 × 10^7^ cells of human PBMCs were quickly injected into a lateral vein of the mouse tail. The human PBMCs used in this study were obtained from a single healthy donor, and characteristics such as population distribution, human leukocyte antigen type, and viability were verified. Five days post-transplantation of human PBMCs, A2058 tumor cells in PBS (2 × 10^6^ cells/100 μL/mouse) were subcutaneously xenografted in the right flanks of NSG-B2m mice (this is day 0 of the experiment).

### 2.6. ngTASO and Anti-PD-1 Antibody Treatment

ngTASO was provided by Autotelic Bio. Inc., and the anti-PD-1 antibody (Pembrolizumab) was purchased from Merck Sharp & Dohme Corp. (Kenilworth, NJ, USA). From day 5 of the experiment, ngTASO (30 mg/kg) and anti-PD-1 antibody (100 μg/mouse) were intraperitoneally administered twice a week and thrice a week, respectively. Tumor volumes were measured using an electronic caliper and estimated by the formula: (distance × width^2^)/2. Relative tumor growth was calculated using the formula: (tumor volume on the day of measurement)/(tumor volume on the day of first administration). Mice were sacrificed on day 23, and xenografted tumors were harvested, weighed, and processed for analysis.

### 2.7. Fluorescence-Activated Cell Sorting (FACS) Analysis

For FACS analysis of human T cell subsets in the blood and xenografted tumor of HIS mice, PerCP/Cyaninne5.5-labeled anti-human CD45 antibody (clone 2D1), Brilliant Violet (BV) 421-labeled anti-mouse CD45 antibody (clone 30-F11), PE/Cyanine7-labeled anti-human CD3 antibody (clone HIT8a), PE-labeled anti-human CD4 antibody (clone SK3), APC/Cyanine7-labeled anti-human CD8a (clone 2D1), BV 650-labeled anti-human CD279 antibody, APC-labeled anti-human CD25 antibody (clone BC 96), and FITC-labeled anti-human Foxp3 antibody (clone 206D) were all procured from BioLegend (BioLegend, San Diego, CA, USA). To analyze the human CTLs and Tregs subsets in the blood of HIS mice, approximately 100 μL of mouse blood was collected by retro-orbital bleeding on day 21. For evaluation of tumor-infiltrating lymphocytes (TILs) in xenografted tumors of HIS mice, 1–2 mm^3^ of tumors harvested on day 23 were minced and incubated at 37 °C for 1 h with a digestion cocktail containing 1 mg/mL collagenase type IV, 0.5 U/mL hyaluronidase type V, and 1 U/mL DNAse I. Digested tumors were passed through a 70 μm cell strainer, and the filtrate was used as a sample of TILs. Red blood cells (RBCs) in the blood and tumor samples were removed by incubation with RBC lysis buffer (BioLegend), followed by incubation with anti-CD16/32 for 10 min. The cells obtained were stained with antibodies labeled with the appropriate fluorochromes for human T cell subsets present in the blood and xenografted tumors of HIS mice. The intracellular staining for Foxp3 was performed using the True-Nuclear^TM^ Transcription factor buffer set (BioLegend). All flow cytometry data were acquired on FACS Symphony A3 (BD Bioscience, San Diego, CA, USA), and data were analyzed using the FlowJo software (Tree Star, San Carlos, CA, USA). 

### 2.8. Immunohistochemistry (IHC)

Harvested xenografted tumors were fixed in 10% neutral buffered formalin and embedded in paraffin. The tissue blocks were then cut into 4 μm slices, collected on slides, and subsequently deparaffinized. After rehydration, antigen retrieval was performed by incubating the slides with 10 mM sodium citrate buffer (pH 6.0) at 100 °C for 10 min. Non-specific responses were reduced by incubation with 3% hydrogen peroxide and 5% BSA. Tissue sections were subsequently incubated overnight with primary antibodies against active β-catenin (clone D12A1; Cell Signaling Technology, Inc., Danvers, MA, USA; 1:200), human PD-L1 (clone E1L3N; Cell Signaling Technology, Inc.; 1:100), human CD8 (clone D8A8Y; Cell Signaling Technology, Inc.; 1:100), human Foxp3 (clone D2W8E; Cell Signaling Technology, Inc.; 1:200), and granzyme B (clone D6E9W; Cell Signaling Technology, Inc.; 1:200). The probed slides were reacted with a biotinylated secondary antibody for 1 h and avidin-biotin peroxidase complexes (ABC Elite kit; Vector Labs, Burlingame, CA, USA) for 30 min. The peroxidase activity was visualized using a DAB kit (Vector Labs), followed by a counter stain with hematoxylin. Images were acquired in at least four fields per slide by the OlyVIA 3.2 software (Olympus, Tokyo, Japan), followed by slide scanning by SLIDEVIEW VS200 digital scanner device (Olympus). The DAB intensity was quantified using the ImageJ Fiji software, as described previously [25], and the number of DAB-positive cells were counted by two experienced researchers in a blind manner.

### 2.9. Statistical Analysis

Statistical significances of the data were analyzed by Student’s *t*-test or one-way analysis of variance (ANOVA) followed by a post hoc Dunnett’s multiple comparison test using the GraphPad Prism 5.01 software (GraphPad Software Inc., San Diego, CA, USA). The results are presented as means ± standard errors of the mean (S.E.M.) or standard deviation (S.D.). The *p*-values < 0.05 are considered statistically significant. 

## 3. Results

### 3.1. ngTASO Downregulates TGF-β2 and Its Downstream Signaling

ngTASO is complementary to specific 20-nucleotide sequences of the human TGF-β2 mRNA and is involved in inhibiting TGF-β2 production. In this study, we observed that ngTASO reduces the TGF-β2 mRNA level at 130.6 nM of half-maximal inhibitory concentration (IC_50_) in A2058 cells (Figure 1A). Western blot was performed to investigate the mechanism of ngTASO on alteration of TGF-β-associated signaling, including expressions of TGF-β ligands (TGF-β1 and TGF-β2), the canonical pathway (p-SMAD2/3/SMAD2/3), and non-canonical pathway (PI3K/Akt signaling). As shown in Figure 1B,C, the TGF-β-associated proteins tend to decrease after exposure to ngTASO. Notably, the levels of TGF-β2 and p-GSK-3β/GSK-3β were significantly decreased by ngTASO treatment. Evaluating the amounts of TGF-β2 in the A2058 cell supernatant using the enzyme-linked immunosorbent assay revealed very low levels, below the detection limit. Furthermore, the serum levels of TGF-β1 and TGF-β2 were significantly decreased in the ngTASO-treated group compared to the vehicle-treated group in melanoma-bearing HIS mice (Appendix A). 

### 3.2. TGF-β2 Blockade by ngTASO in Combination with PD-1 Inhibition Delays Tumor Growth

To investigate the antitumor immune interaction between human melanoma and human T cells, we generated melanoma-bearing humanized mice by transplanting human PBMCs followed by inoculation of A2058 cells, into NSG-B2m mice (Figure 2A). Tumor volume was checked thrice a week after the administration of ngTASO and/or anti-PD-1 antibody. We observed that compared to the PBMC+vehicle group, co-treatment with ngTASO and anti-PD-1 antibody significantly delayed the tumor growth from day 6 post-administration (Figure 2B). Tumor weights were measured after the sacrifice of the animals (day 23). Similar to the tumor volume results, administration of ngTASO in combination with anti-PD-1 showed significantly decreased tumor weights compared to the vehicle-treated group (Figure 2C). These results imply that ngTASO combined with anti-PD-1 exerts an additional effect on growth inhibition of xenografted A2058 tumors.

### 3.3. ngTASO and Anti-PD-1 Administration Modifies the Subpopulation of Human CD8+ T Cells and Tregs in Peripheral Blood

Since the anticancer immune activity in the body is systemically affected by the difference in the composition of T cell subsets, we determined the proportion of CTLs and Tregs in peripheral blood by FACS analysis. Treatment of ngTASO and/or anti-PD-1 presented no significant alteration in human CD8+ cells among the human CD3+ cells in peripheral blood (Figure 3A,C). However, the percentage of human PD-1+ cells in the human CD8+ cell population in the peripheral blood was significantly reduced after administering the anti-PD-1 antibody (Figure 3A,D). As described in a previous study [26], decreased detection of PD-1+ cells by flow cytometry might result from a competitive binding to its target between anti-PD-1 therapy and subsequent anti-PD-1 staining for FACS. The proportion of human CD25+Foxp3+ cells among human CD4+ cells was significantly decreased in the ngTASO treated groups as compared to the PBMC+vehicle group (Figure 3B,E). These results indicate that the inhibition of TGF-β2 by ngTASO in combination with PD-1 mediates the downregulation of inhibitory T cell subsets in peripheral blood.

### 3.4. TGF-β2 Blockade by ngTASO Combined with PD-1 Inhibition Downregulates the β-Catenin Activation and Facilitates Human CD8+ T Cell Infiltration in Tumors

Due to its significance in T-cell exclusion and resistance to anti-PD-1 antibody therapy, we evaluated the degree of tumor-intrinsic active β-catenin expression [27,28]. As presented in Figure 4A,C, we observed significantly decreased expression in active β-catenin in the anti-PD-1 alone-treated group or ngTASO and anti-PD-1 co-treated group compared to the vehicle-treated group. As altered systemic and local microenvironments, such as immune composition and biomarkers, affect the TIL levels [29], we next examined the number of infiltrated human CD45+ cells and the ratio of human CD8+ T cells to human CD4+ T cells in the xenografted tumor. Compared to the vehicle-treated group, co-treatment with the ngTASO and anti-PD-1 antibody showed a significant infiltration of human CD45+ cells into the xenografted A2058 tumor (Figure 4D). Moreover, the ratio of human CD8+ T cells to human CD4+ T cells was significantly increased in the ngTASO and anti-PD-1 co-treated groups compared to the vehicle-treated group (Figure 4B,E).

### 3.5. TGF-β2 Blockade by ngTASO Combined with PD-1 Inhibition Upregulates PD-L1 Expression in Tumors

Since the exhaustion of TILs is largely mediated by the PD-L1/PD-1 axis [30], we evaluated the PD-L1 expression of xenografted tumors and the PD-1 expression in tumor infiltrated CTLs. As shown in Figure 5A,C, PD-L1 expression in the tumor was notably upregulated by the TGF-β2 blockade in combination with PD-1 inhibition. In addition, the percentage of human PD-1+ cells among the human CD8+ cells in xenografted tumors was significantly diminished in the anti-PD-1 treated groups (Figure 5B,D). 

### 3.6. TGF-β2 Blockade by ngTASO Combined with PD-1 Inhibition Regulates Infiltration of the T Cell Subpopulation and Their Activation in Tumors

Since alterations in the immune cell composition and distribution in tumor sites are closely related to activation of CTLs [31], we evaluated the population and distribution of human CD8+ cells and human Foxp3+ cells in xenografted tumors by performing IHC analysis. As presented in Figure 6A,B, xenografted A2058 tumors in hu-PBL NSG-B2m mice were the immune-excluded type with relatively low levels of human CD8+ cells in the tumor margin. However, exposure to the ngTASO or anti-PD-1 antibody slightly mediated the infiltration of human CD8+ cells into the center of the tumor. In particular, co-treatment of ngTASO and anti-PD-1 significantly increased the human CD8+ cell infiltration throughout entire tumors. Moreover, although only small numbers of human Foxp3+ cells were detected in the tumor, ngTASO combined with anti-PD-1 antibody significantly reduced the number of intra-tumoral Foxp3+ cells (Figure 6A,C). Since activated CTLs secrete granzyme B to induce a cytotoxic effect on tumor cells [32], we also examined the expression levels of granzyme B in the xenografted tumor by IHC. Correlating with the results of the human CD8+ cell and human Foxp3+ cell expression, the granzyme expression was significantly increased after co-treatment with ngTASO and anti-PD-1, as compared with vehicle treatment (Figure 6A,D).

## 4. Discussion

TGF-β is commonly known as an immunosuppressive cytokine and plays a primary role in the immunological regulation of the microenvironment in various cancers, including breast cancer, melanoma, and pancreatic cancer [33,34]. The therapeutic approaches to target TGF-β signaling for cancer include the monoclonal antibody against TGF-β ligand, receptor kinase inhibitors, vaccines, and ASO. However, TGF-β inhibitors have not been approved due to their limited clinical efficacy [35]. ASO technology is one of the most promising nucleic acid-based therapeutic approaches, imparting low toxicity due to its design that targets the gene sequence alone. To date, TGF-β2 ASO (also known as Trabedersen) has been clinically developed, but its therapeutic efficacy has not been remarkable (NCT00761280 and NCT00844064). ngTASO, a novel next-generation ASO of TGF-β2 inhibition, was designed to improve the mRNA binding affinity via an optimized ASO sequence screening method and enhance plasma stability via modification of the molecular structure of sugar. In this study, TGF-β2 production in A2058 cells was significantly downregulated after exposure to ngTASO, and the blockade of TGF-β signaling by ngTASO combined with PD-1 inhibition showed additional tumor growth inhibition in a melanoma-bearing HIS mouse model. 

Downregulation of TGF-β signaling has been demonstrated to switch the tumor phenotype from immune-excluded to immune-inflamed [36]. In the immune-excluded melanoma-bearing HIS mouse model, we determined that the infiltration of CTLs into the tumor site is augmented by the blockade of TGF-β2 combined with PD-1 inhibition. The TGF-β signaling pathway is closely related to the Wingless-related integrin site (Wnt)/β-catenin signaling pathway in melanoma, and TGF-β-mediated stromal fibrosis is closely associated with the activation of tumor-intrinsic β-catenin [37,38]. The Wnt/β-catenin signaling pathway is one of the primary regulators of T cell exclusion from the TME in many cancers, including hepatocellular carcinoma and melanoma [27,28,39]. In the current study, we confirmed that the expression of intrinsic active β-catenin is correlated with the expression of TGF-β2 levels in A2058 cells, and the infiltration of CTLs into melanoma might be partly hampered by the tumor intrinsic active β-catenin. Since ngTASO exclusively acts on the human TGF-β2 mRNA sequence, it is considered that downregulation of the tumor-derived TGF-β2 by ngTASO might primarily contribute to the reduction of intrinsic active β-catenin in xenografted A2058 tumors. As antitumor effects of CTLs by ICIs are mainly mediated by a contact-dependent mechanism [6], the immunotherapeutic efficacy of anti-PD-1 monotherapy is poor in immune-excluded A2058 tumors. However, the effects of anti-PD-1 are enhanced by co-treatment with ngTASO by facilitating the infiltration of CTLs throughout the entire tumor, partly mediated by downregulation of the intrinsic active β-catenin and improving the T cell-mediated antitumor immunity, resulting in the enhanced delay of tumor growth. 

Tumors escape from the immune surveillance by upregulating the TGF-β signaling derived from systemic immunosuppression as well as local immunoediting in TME [33,34]. It has been demonstrated that the tumor-derived TGF-β induced extrathymic conversion of naïve CD4+ T cells induces Foxp3+ Tregs [21,40]. Consistent with previous studies conducted in vitro or in syngeneic mouse models [4,21], we observed that the blockade of TGF-β signaling by ngTASO diminishes the Tregs constitution in the peripheral blood of hu-PBL NSG-B2m mice. Tregs downregulate the antitumor immunity by producing anti-inflammatory cytokines, including IL-10, IL-35, and TGF-β, and suppressing effector cells, such as CTLs and NK cells [41,42]. The current results show that the blockade of TGF-β2 probably contributes to a systemic immunostimulatory state by downregulation of Tregs. Previously, we showed that the blockade of TGF-β2 reduces the intratumoral ratio of Tregs to CTLs in the immune-inflamed model [22]. However, in the current study, there is no remarkable downregulation of Tregs by TGF-β2 inhibition alone due to the low baseline frequency of Tregs in the immune-excluded tumors. However, TGF-β2 blockade in combination with PD-1 inhibition significantly downregulated the ratio of Tregs to CTLs, and CD4+ cells to CD8+ cells, at the tumor site. Intratumoral Tregs mediate a tumor-promoting microenvironment, suppress the tumor-killing potential of CTLs, and are associated with poor prognosis in many cancers [31,42,43]. Therefore, a downregulated ratio of Tregs to CTLs at the tumor site could probably mediate increased granzyme B secretion and consequently enhance the T cell-mediated tumor growth inhibition.

It has previously been demonstrated that the immunological state in the TME largely determines the therapeutic benefit from ICIs [29,44]. Several studies revealed that patients with pre-existing TILs present an enhanced response rate to ICIs, and increased infiltration of CTLs in tumors is associated with a good prognosis [45,46]. Despite the growing success of ICIs in the treatment of advanced melanoma, a subset of melanoma patients with low immunogenicity in the tumor are resistant to ICIs [44,47]. Overcoming this limitation of anti-PD-1 treatment of malignant melanomas is one of the main concerns in immuno-oncology. To this effect, various therapeutic strategies to convert immunologic subtypes from a cold tumor (immune-desert or immune-excluded) to a hot tumor (immune-inflamed) are being researched. The blockade of TGF-β signaling facilitates the infiltration of CTLs in the tumor, but the persistent suppression of TGF-β signaling mediates PD-L1 expression in the tumor via increased secretion of IFN-γ and TNF-α from TILs [48,49]. In the current study, we observed upregulated expression of human PD-L1 in tumors by TGF-β2 blockade combined with PD-1 inhibition, which is associated with increased TILs. Although PD-L1 expression in tumors facilitates immune escapes, tumors with PD-L1 negative expression present a worse response to PD-1/PD-L1 blockade therapies [49,50]. Concomitant inhibition of TGF-β receptor I and PD-1 synergistically represses tumor promotion in a genetically engineered mouse model of pancreatic ductal adenocarcinoma, and clinical studies using a combination of TGF-β blockers and PD-1/PD-L1 inhibitors are currently underway [35,48]. Results from the current study indicate that TGF-β2 inhibition by ngTASO might improve the response to anti-PD-1 immunotherapy by upregulating PD-L1 expression with increased TILs in immune-excluded melanoma. As anti-PD-1 immunotherapy mediates antitumor immunity via antigen-specific T cell responses, a combination of anti-PD-1 with TGF-β2 inhibition, which might mediate antigen non-specific T cell responses as well as antigen-specific T cell responses, may have additional advantages in the development of immunotherapy. 

## 5. Conclusions

In conclusion, we determined that TGF-β2 inhibition by ngTASO enhances the T cell-mediated antitumor immunity of anti-PD-1 therapy by facilitating the intratumoral infiltration of CTLs, is partly associated with the downregulation of intrinsic β-catenin crosstalk, and consequently delays tumor growth in a melanoma-bearing HIS mouse model. In addition, TGF-β2 blockade in combination with PD-1 inhibition might contribute to the formation of an immunostimulatory status in systemic and TME by mediating suppression of Tregs and counteracting compensatory increases in the PD-L1 expression in tumors. These results indicate that the combination of ngTASO and anti-PD-1 has an enhanced potential in T cell-mediated antitumor immunity. This combination therapy is, therefore, a potentially promising strategy for immune-excluded melanoma.

## Figures and Tables

**Figure 1 cancers-14-05220-f001:**
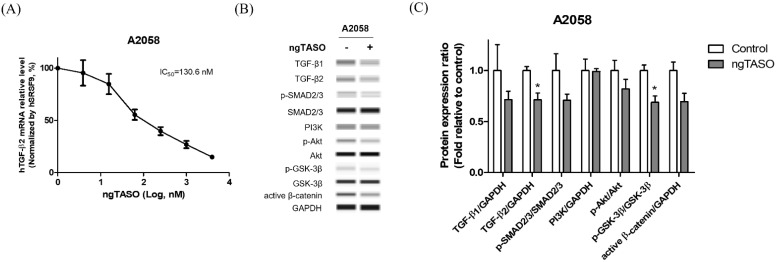
Effects of ngTASO on TGF-β2 production and its associated downstream signaling in A2058 cells. Alteration of TGF-β-associated downstream signaling was measured. (**A**) TGF-β2 mRNA expression levels were measured by qRT-PCR and normalized to *SRSF9* expression. (**B**) Representative Western blot band images and (**C**) relative intensity ratios of each protein expression in A2058 were presented. Data are expressed as means ± S.D. from triplicate wells. * *p* < 0.05 vs. control group (Student *t*-test).

**Figure 2 cancers-14-05220-f002:**
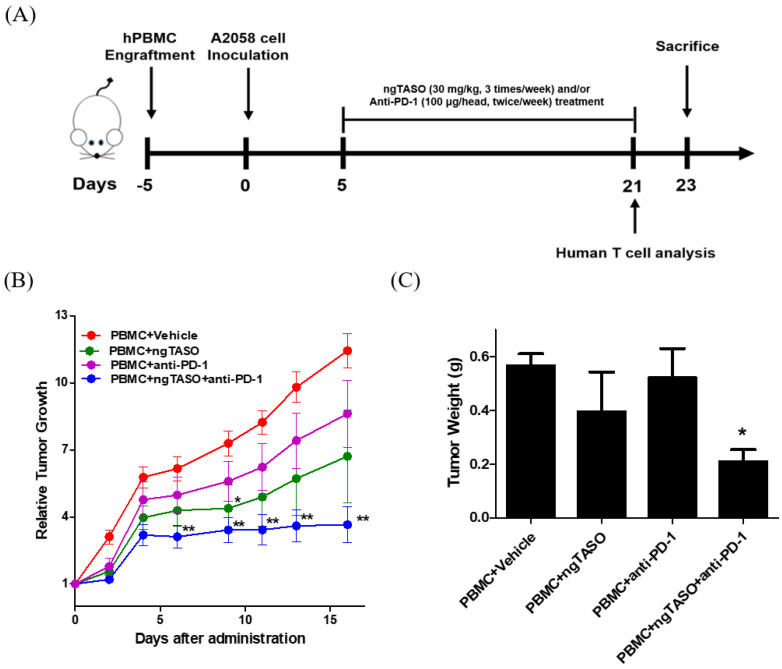
Effects of ngTASO and anti-PD-1 co-treatment on tumor growth. Melanoma-bearing human immune system mouse model was generated by transplanting human PBMC followed by inoculation of A2058 cells into NSG-B2m mice. (**A**) Experimental scheme for the establishment of melanoma-bearing hu-PBL NSG-B2m mice and treatment of ngTASO and/or anti-PD-1 antibody. (**B**) Relative tumor growth was calculated using the formula ((tumor volume on the day of measurement)/(tumor volume on the day of first administration)). (**C**) Tumor weights were measured at study termination (23 days after tumor inoculation). The results are expressed as mean ± S.E.M. obtained from six mice per group. * *p* < 0.05 and ** *p* < 0.01 vs. PBMC+Vehicle group (Dunnet’s test).

**Figure 3 cancers-14-05220-f003:**
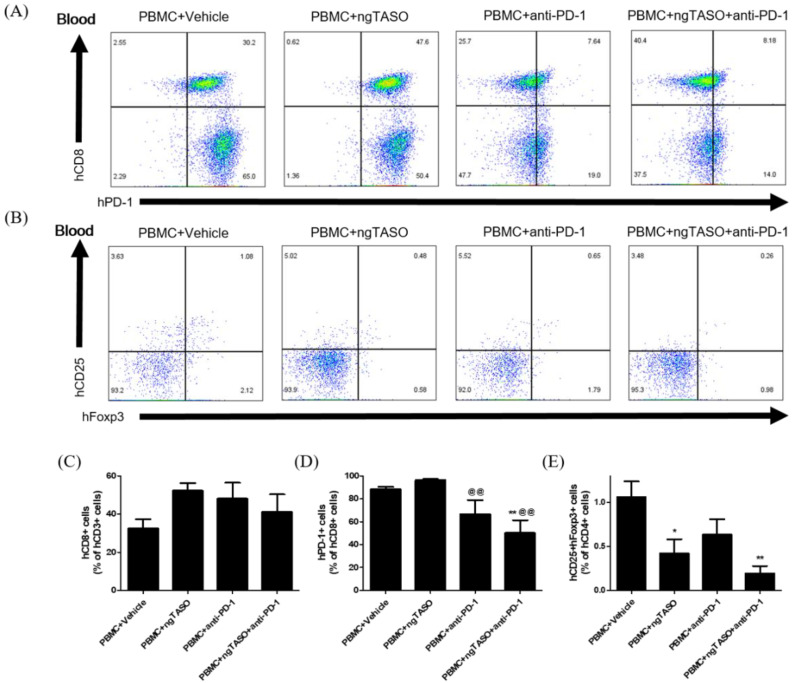
Effects of ngTASO and anti-PD-1 co-treatment on reconstitution of T cell subsets in peripheral blood. Subpopulations of human T cells in peripheral blood of hu-PBL NSG-B2m mice were measured by FACS analysis on day 21. (**A**) The representative flow cytometry plots of human CTLs (hCD8+PD-1+ cells) in the hCD3+-gated population and (**B**) human Tregs (hCD25+hFoxp3+ cells) in the hCD4+-gated population in peripheral blood of hu-PBL NSG-B2m mice. Data showing the percentage of (**C**) hCD8+ cells among hCD3+ cells, (**D**) hPD1+ cells among hCD8+ cells, and (**E**) hCD25+hFoxp3+ cells among hCD4+ cells. The results are expressed as mean ± S.E.M. obtained from five to six mice per group. * *p* < 0.05 and ** *p* < 0.01 vs. PBMC+Vehicle group; ^@@^
*p* < 0.01 vs. PBMC+ngTASO group (Dunnet’s test). h, human.

**Figure 4 cancers-14-05220-f004:**
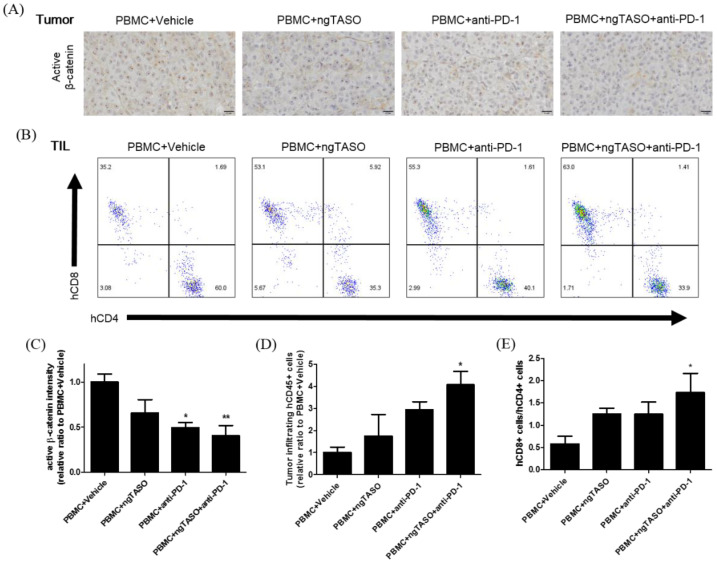
Effects of TGF-β2 blockades combined with PD-1 inhibition on activation of β-catenin and infiltration of CD8+ cytotoxic T lymphocytes (CTLs). Active β-catenin expression level in the xenografted tumor was measured by IHC, and the subpopulation of tumor infiltrated human T cells was assessed by FACS analysis. (**A**) Representative IHC images against active β-catenin, and (**B**) representative flow cytometry plot of hCD4+ and hCD8+ cells in an hCD3+-gated population. (**C**) The relative DAB intensity ratio of active β-catenin was evaluated in at least four fields per specimen and five to six specimens per group. The relative ratio of (**D**) tumor-infiltrating hCD45+ cells and (**E**) the ratio of hCD8+ cells to hCD4+ cells among tumor infiltrated hCD3+ cells were presented. FACS data were obtained from five to six mice per group. The results are expressed as mean ± S.E.M. * *p* < 0.05 and ** *p* < 0.01 vs. PBMC+Vehicle group (Dunnet’s test). Bar = 20 μm. h, human.

**Figure 5 cancers-14-05220-f005:**
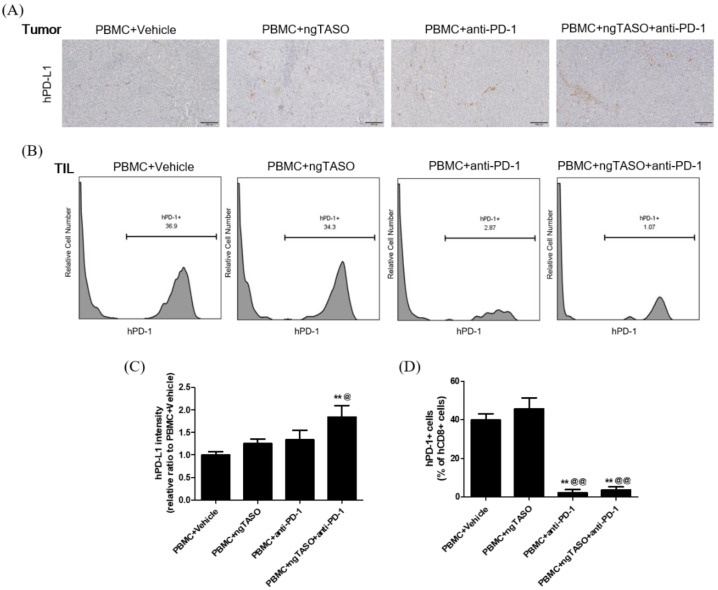
Effects of TGF-β2 blockades combined with PD-1 inhibition on PD-L1 expression and subpopulation of PD-1+CD8+ cells in the tumor. The expression level of human PD-L1 in xenografted tumors was evaluated by IHC and the subpopulation of hPD-1+ cells in tumor infiltrated hCD8+ cells was assessed by FACS analysis. (**A**) Representative IHC images for hPD-L1 in tumors, and (**B**) representative histogram of FACS analysis. (**C**) The relative intensity ratio of hPD-L1 was assessed in at least four fields per specimen and five to six specimens per group. (**D**) The percentage of hPD-1+ cells among hCD8+ cells in xenografted tumors was presented. FACS data were obtained from six mice per group. The results are expressed as mean ± S.E.M. ** *p* < 0.01 vs. PBMC+Vehicle group; ^@^ *p* < 0.05 and ^@@^ *p* < 0.01 vs. PBMC+ngTASO group (Dunnet’s test). Bar = 100 μm. h, human.

**Figure 6 cancers-14-05220-f006:**
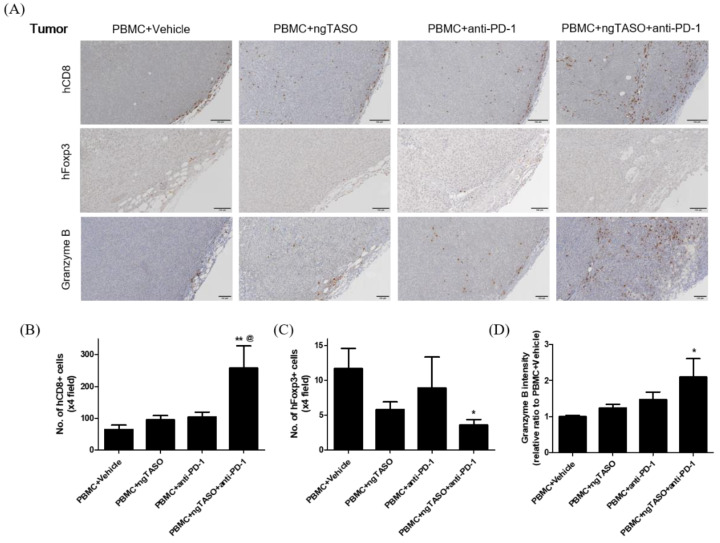
Effects of TGF-β2 blockades combined with PD-1 inhibition on the distribution of tumor infiltrated T cells and their activation. Infiltration and distribution, and activation of human T cells in the tumor were measured by IHC staining against hCD8, hFoxp3, and granzyme B. (**A**) Representative IHC images of hCD8, hFoxp3, and granzyme B expression. The number of (**B**) hCD8+ cells and (**C**) hFoxp3+ cells per ×4 field, and (**D**) the relative DAB intensity ratio of granzyme B were assessed in at least four fields per specimen and five to six specimens per group. The results are expressed as mean±S.E.M. * *p* < 0.05 and ** *p* < 0.01 vs. PBMC+Vehicle group; ^@^ *p* < 0.05 vs. PBMC+ngTASO group (Dunnet’s test). Bar = 100 μm. h, human.

**Table 1 cancers-14-05220-t001:** Primer sequences of target genes used for qRT-PCR.

Target	Species	Forward/Reverse	Sequence (5′-3′)	Tm (°C)
*TGFB2*	Human	F	CAGCACACTCGATATGGACCA	57
		R	CCTCGGGCTCAGGATAGTCT	61
*SRSF9*	Human	F	TGTGCAGAAGGATGGAGT	55
		R	CTGGTGCTTCTCTCAGGATA	54

## Data Availability

The data presented in this study are available on reasonable request from the corresponding author.

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
