# Peer review of "Next-Generation Antisense Oligonucleotide of TGF-β2 Enhances T Cell-Mediated Anticancer Efficacy of Anti-PD-1 Therapy in a Humanized Mouse Model of Immune-Excluded Melanoma"

_cancers, 2022, doi:10.3390/cancers14215220_

Round 1
Reviewer 1 Report
The manuscript titled: “ATB320, a next-generation antisense oligonucleotide of TGF-β2, enhances T cell-mediated anticancer efficacy of anti-PD-1 therapy in a humanized mouse model of immune-excluded melanoma” is well written. The manuscript is based on a well-constructed scientific concept and carried out the studies are well. However, data needs to be refined in a presentable manner. The functional studies of tumor-infiltrating T cells are missing in the manuscript. The present manuscript would be benefited by addressing the points below. I would suggest accepting the manuscript after minor revision.
Comments:
· In figure 3, authors have shown that the expression of PD-1 is effectively downregulated in anti-PD1 and combination (anti-PD-1therapy and ATB320) therapy groups, however, the molecular mechanism behind it needs to be studied.
· In figure 6, authors have shown that TGF-b inhibition with anti-PD-1 therapy promoted T cell sub-population activation markers, to confirm, authors need to do the T cells functional studies such as T cell proliferation assay and /or T cell killing assay to confirm the phenotypes shown in figure 6.
Author Response
Reviewer 1:
The manuscript titled: “ATB320, a next-generation antisense oligonucleotide of TGF-β2, enhances T cell-mediated anticancer efficacy of anti-PD-1 therapy in a humanized mouse model of immune-excluded melanoma” is well written. The manuscript is based on a well-constructed scientific concept and carried out the studies are well. However, data needs to be refined in a presentable manner. The functional studies of tumor-infiltrating T cells are missing in the manuscript. The present manuscript would be benefited by addressing the points below. I would suggest accepting the manuscript after minor revision.
Comments:
In figure 3, authors have shown that the expression of PD-1 is effectively downregulated in anti-PD1 and combination (anti-PD-1therapy and ATB320) therapy groups, however, the molecular mechanism behind it needs to be studied.
--> As mentioned in the results section, the decreased detection of PD-1+ cells by flow cytometry might result from a competitive binding to its target between anti-PD-1 therapy and anti-PD-1 staining for FACS. This is consistent with the previous study we cited (ref. 26) and our previous in-house data (The number of PD-1+ cells gradually decreased over time after anti-PD-1 treatment, which reflected the half-life of anti-PD-1 in mouse blood.). And in the results of PD-1 expression in tumors using IHC, there was no significant decrease in the expression level of PD-1 compared to the CD8+ cell numbers between experimental groups (in-house data, data not shown). However, as the reviewer commented, it is considered that more detailed confirmation of the effect of combination (TGF-b2 inhibition+anti-PD-1) therapy on PD-1 expression and its molecular mechanism needs to be performed by a further study.
In figure 6, authors have shown that TGF-b inhibition with anti-PD-1 therapy promoted T cell sub-population activation markers, to confirm, authors need to do the T cells functional studies such as T cell proliferation assay and /or T cell killing assay to confirm the phenotypes shown in figure 6.
--> As shown in Fig.2, we confirmed the effect of combination (TGF-b2 inhibition+anti-PD-1) therapy on T cell reconstitution in blood. These results can be considered to reflect the effect of combination (TGF-b2 inhibition+anti-PD-1) therapy on T cell proliferation, and changes in the subpopulations of CD8 and Treg were also confirmed. By replacing the study on the T cell killing assay mentioned by the reviewer, in vitro LAK (lymphokine-activated killer cell)- induced cell(A2058) death assay was performed. Cell viability in the TGF-b2 inhibitor-treated group was approximately 90% of the control, and it was not significant (data not shown). If the reviewer meant functional studies using TILs, the experiment could not proceed due to the difficulty of obtaining sufficient numbers of TILs from xenografted A2058 tumors in human immune system mice. However, we believe that one of the main causes of suppressing the effects of existing anti-PD-1 is the immune excluded environment, so it can be used as a combination therapy by inhibiting the TGF-b pathway to affect the conversion to the immunogenic environment using a concept of ASO.
Reviewer 2 Report
1. In line Line 33 the authors use the term "immune excluded" .Why are the tumors immune excluded?I dont relate to this term. The tumors have immune subsets either functional or exhausted. What are the authors trying to imply by the use of this term?
2. A better representative wester blot should be included in the Fig1B. The current image is not convincing enough.
3. In fig4A , decreased expression if beta-catenin was observed in which cells or in the tumors? This observation is not evident in the representative IHC staining shown in the manuscript.
4. For fig 4D is there a flow plot associated with quantification of CD45.1 infiltration in the TILS?
5. In fig. 4B the flow plots looks a bit overcompensated. This may affect the percentages of cell subsets. Particularly since the authors use the ratio.
6. The quantification of FOXP3 positive CD4 cells in the TIL would be nice to see since TFG-beta is being targeted. Similar to what is shown in the peripheral blood.
7.In fig 5A and C what is the explanation for elevated levels of PD-L1 in the tumor cells? Is this an expected outcome?
8. Flow cytometry analysis of the TIL for immune pheno-typing would have greatly supported the conclusions of this manuscript.

Author Response
Reviewer 2:
1. In line Line 33 the authors use the term "immune excluded" .Why are the tumors immune excluded?I dont relate to this term. The tumors have immune subsets either functional or exhausted. What are the authors trying to imply by the use of this term?
--> Solid tumors are commonly classified according to the frequencies of immune cells in the tumor (immune-inflamed, immune excluded, and immune deserted), and in CDX in human immune system mice, the immunophenotype is different depending on the cell line. In the case of A2058, it is confirmed that hCD8+ cells are lined along the edge of the tumor when no test substance is treated. Also, the number is relatively small. This is considered a form of immune exclusion. On the other hand, when the TGF-b2 inhibition+anti-PD-1 was treated, more hCD8 cells penetrated throughout the tumor including the central zone of the tumor, and appeared in a more immunogenic form. Especially, melanoma is commonly known to be an immunogenic type, but depending on the patient, there are cases where it is less immunogenic, and in this case, it is known that the case does not respond well to immunotherapy. So, efforts are being made to solve this point. As part of that, we used ASO, which inhibits TGF b2 production.
2. A better representative wester blot should be included in the Fig 1B. The current image is not convincing enough.
--> The results of Western blot was derived from using an automated system (capillary-based WB device), and the blot image can be adjusted through the processing. The authors have slightly modified that image as shown.
3. In fig 4A, decreased expression if beta-catenin was observed in which cells or in the tumors? This observation is not evident in the representative IHC staining shown in the manuscript.
--> We used an activated b-catenin antibody (Non-phospho b-catenin). Commonly, active b-catenin translocated to the nucleus (also found in cytoplasmic and membrane), and DAB reaction (Brown color) is observed in the center of cells condensed (Fig 4A). We have semi-quantified this DAB intensity value (protein expression, brown), which was normalized by the nuclear intensity (hematoxylin, blue, ref 25) and presented its relative values.
4. For fig 4D is there a flow plot associated with quantification of CD45.1 infiltration in the TILS?
--> Of course, we have the .fcs file, and also checked the flow plot for hCD45+ cells. However, the proportion of TIL in the entire tumor tissue was very low, so the flow plot was not suitable for showing and comparing with each group. The tumor-infiltrated hCD45 result in Fig.4D was presented as a relative ratio to the PBMC+Vehicle group, and the proportion of tumor-infiltrated hCD45 cells compared to the total cells in the PBMC+Vehicle group was about 0.1%.
5. In fig. 4B the flow plots looks a bit overcompensated. This may affect the percentages of cell subsets. Particularly since the authors use the ratio.
--> We performed compensation for all fluorescence-labeled antibodies and applied this to the program to auto-compensate by itself. We have modified the flow plot and reanalyzed the data. As you mentioned, the percentages of cell subsets were slightly changed (No event is on the edge of the plot chart.).
6. The quantification of FOXP3 positive CD4 cells in the TIL would be nice to see since TFG-beta is being targeted. Similar to what is shown in the peripheral blood.
--> As the reviewer mentioned, it seems like a good way to do it, and we actually did it. However, the human immune system mouse model employed in this study has a limited amount of samples and it is difficult to obtain a sufficient number of lymphocytes from the xenografted tumor, especially from the immune-excluded sample. As mentioned in the discussion section, there is no remarkable downregulation of Foxp3+ hTregs by TGF-b2 inhibition due to the very low frequency of hTregs in the control group (xenografted A2058 showed immune excluded phenotype in this study). Thus, the number of Tregs(hCD4+hCD25+Foxp3+ cells) acquired was insufficient to analyze the proportion of Tregs in TIL by FACS. However, alternatively, through IHC analysis, we counted Foxp3+ cells in tumor tissue, and it was found that Treg was reduced by the combination therapy (TGFb2 inhibition+anti-PD-1), similar to results in the peripheral blood.
7. In fig 5A and C what is the explanation for elevated levels of PD-L1 in the tumor cells? Is this an expected outcome?
--> Actually, the elevated PD-L1 in the combination therapy group (TGF-b2 inhibition+anti-PD-1) was a result that could be inferred to some extent, suggesting the possibility or advantage of this combination strategy. We mentioned this point in the discussion section. (Blockade of TGF-β signaling facilitates the infiltration of CTLs in the tumor, but the persistent suppression of TGF-β signaling mediates PD-L1 expression in the tumor via increased secretion of IFN-γ and TNF-α from TILs) Currently, PD-1/PD-L1 immune checkpoint inhibitors are considered the most important therapeutic intervention in immune-oncology, and one of the major shortcomings of current ICI may be the lack of response in certain cancer including melanoma attributed to its low immunogenicity. At this point, since increased PD-L1 expression in melanoma can be also associated with increased pre-existed TILs by the combination of TGFb2 inhibition and anti-PD-1, this means that it has the potential to increase the response of existing anti-PD-1 therapy.
8. Flow cytometry analysis of the TIL for immune pheno-typing would have greatly supported the conclusions of this manuscript.
--> The authors really appreciate for reviewer’s comments to complete this study.
Round 2
Reviewer 1 Report
The manuscript titled: “ATB320, a next-generation antisense oligonucleotide of TGF-β2, enhances T cell-mediated anticancer efficacy of anti-PD-1 therapy in a humanized mouse model of immune-excluded melanoma” is well written and revised. The manuscript is based on a well-constructed scientific concept and carried out the studies are very well. The authors have addressed the comments up to their level. I would like to accept the manuscript in its current form.